# Study on Modification and Mechanism of Construction Waste to Solidified Silt

**DOI:** 10.3390/ma16072780

**Published:** 2023-03-30

**Authors:** Yannan Shi, Haoxuan Weng, Jiongqi Yu, Yongfan Gong

**Affiliations:** 1Zhejiang Guangchuan Engineering Consulting Co., Ltd., Hangzhou 310020, China; 2Zhejiang Provincial Key Laboratory of Water Resources Disaster Prevention and Mitigation, Hangzhou 310020, China; 3College of Architectural Science and Engineering, Yangzhou University, Yangzhou 225127, China

**Keywords:** dredged sediment, skeleton structure, curing agent, mechanism analysis, stability

## Abstract

A large amount of silt may be produced in river and lake regulation. It not only occupies land but also pollutes the environment. Therefore, it is urgent to seek effective disposal and utilization methods. Based on the problems of poor stability of stabilized soil and its tendency to soften easily in water, as well as its low strength with low curing agent dosage, this paper proposes a method to improve stabilized soil’s solidification effect by adding materials such as cement, lime, fly ash, triethanolamine, sodium hydroxide, sodium silicate, etc., while mixing different grain diameters and quantities of building waste materials and ordinary sand. Using construction waste and ordinary sand as a comparative test, the curing mechanism of construction waste debris on the mechanical properties, permeability, and microstructure of solidified sludge was studied through unconfined compression tests, dry and wet cycle tests, permeability tests, and micro-structure tests such as XRD, MIP, and SEM. The test results show that the strength increases 8.5%~72.1% by adding building waste materials, and it grew with the increase in particle size and amount. It reduced the content of large pore size of solidified sediment and optimized the internal pore structure. At the same time, it formed a new structure filled by rigid skeleton material. Thus, it improved its unit section stress, built up the curing effect and water stability. The findings of this study can be used to modify solidified silt to improve stability and compaction characteristics.

## 1. Introduction

The regulation of rivers and lakes is often inseparable from sludge dredging, which is often accompanied by problems such as large sludge volume, high water content and difficult disposal. How to achieve scientific reduction, rational utilization and green environmental protection has become a hot topic in research [1,2,3]. For example, the Pearl River Delta generates 80 million m^3^ of dredged sludge every year. During the 13th Five-Year Plan period (2016–2020) in Zhejiang Province, the dredging sludge exceeded 360 million m^3^ [4,5,6]. Generally, solidified sludge is a kind of ecological environmental material, which can be used in engineering filling, ecological slope stabilization and other areas. It has the advantages of rich resources, simple preparation, recycling and so on [7,8,9].

Researchers from various disciplines and fields have carried out a lot of research on the influence of factors such as moisture content of solidified soil, curing agent, organic matter and pH on the strength and the modification of solidified soil [10,11,12,13,14]. Some researchers [15,16] have mixed organic modifiers such as plant fiber and starch into the Yellow River silt soil to improve soil cohesiveness. They added polymer short fiber into solidified sludge for strength replenishment. Other researchers [17,18] modified the solidified clay using steel waste slag, slag and other wastes and found that the modified solidified clay has high strength and water resistance. Most of the researchers [19] have used cement, slag, fly, ash and gypsum as the curing agent and cooperated with polypropylene fiber to improve the sludge-stabilized soil. Xu [20,21,22] proposed the development of new empirical prediction models to evaluate swell pressure and unconfined compression strength of expansive soils. The research results can help researchers readily evaluate the swell-strength characteristics of the widespread expansive soils and the sustainable use of construction wastes. The compressive and tensile properties were significantly improved, but the permeability was not significantly improved. At present, the solidified soil still has poor water stability and is easy to disintegrate in the dry–wet cycle, which is difficult to meet the long-term deformation and stability requirements in complex environments.

“Treating waste with waste” is a current hot topic for research in environmental protection and treatment. Although industrial solid waste has been proved to have a good effect on the modification of solidified soil, the properties of solid waste materials are uneven, the performance is unstable, and the chemical and physical properties of river sludge vary greatly. How to use them more efficiently is still a current problem. Therefore, in order to solve the stability of the solidified soil and improve the mechanical properties, this paper proposes to replace part or all of the building waste debris such as sintered brick slag and waste concrete wall as the skeleton matrix of the solidified dredging sludge material. By controlling the amount of admixture and particle size, the strength of the solidified sludge and solidification effect will be adjusted. X-ray diffraction, and scanning electron microscopy were used to analyze the physical phase, pore structure and micro morphology of the solidified sludge. This analysis will reveal the mechanism of action and solidification of construction waste debris in the process of solidification of dredged sludge, and screen out the better scheme. Finally, a theoretical basis for the application of solidified soil in water conservancy, municipal, construction and other industrial projects will be provided. It is therefore recommended that construction waste debris can be reliably used for modifying solidified silt to improve stability and compaction characteristics.

## 2. Materials and Methods

### 2.1. Raw Materials

In this study, the dredged sludge of lake was selected as the test sludge. The basic physical and chemical properties and particle composition of the test sludge were measured according to the Geo-technical Test Specification (SL 237-1999). The water content was about 97.6%, the liquid limit was 53.2% and the plastic limit was 28.3%. Among them, the content of soil particles with a particle size of less than 0.005 mm was about 32.7% and the content of soil particles with a particle size of 0.005 mm~0.075 mm was about 67.3%, which was clay. According to the characteristics of test sludge, P·II 52.5 Portland cement (C), lime (CaO), fly ash (FA), triethanolamine (TEA), sodium hydroxide (NaOH) and sodium silicate (Na_2_SiO_3_) were used as the curing materials, with the mass ratio of C:CaO:FA:TEA:NaOH:Na_2_SiO_3_ = 27:11.85:39:0.15:3:19 and the mixing amount of the curing materials 8% of the dry soil mass, marked as “G8”. “G0” was not mixed with any reagent. In this test, building waste (S), such as sintered brick slag and concrete wall slag, was used to fill the pores of solidified sludge. The building waste was mainly of a small particle size, which was 8.63% below 0.075 mm, 78.86% below 0.075 mm and 12.51% above 2 mm. Single graded sand (S’) with particle sizes of 0.075~0.25 mm, 0.25~0.5 mm, 0.5~1 mm and 1~2 mm was mixed in, and ordinary river sand (S’) with the same particle size distribution was used as the control test.

### 2.2. Methods

#### 2.2.1. Unconfined Compressive Strength Test

Take 13 equal parts of sludge of a certain quality and remove impurities. Add different proportions of solidified materials, construction waste fragments or ordinary river sand according to the quality of dry soil. The proportions are shown in Table 1. With reference to the heavy compaction test in the Soil Test Specification (SL 237-1999), the unit volume compaction work was 2684.9 kJ/m^3^; therefore, use a small three valve compaction cylinder with a diameter of 37.5 mm and a height of 76.4 mm to fully mix the mixture. and air dry it until the moisture content is reached 40%. The compaction cylinder is evenly added in three layers. Each layer is compacted with 100 free drops by the compaction hammer. After each layer is compacted, roughen it with a Geo-technical knife, and then add the next layer of soil sample. After compaction, use a scraper to smooth the surface of the sample, wrap it with a sealing film, and place it in a standard curing box with a temperature of 20 ± 2 °C and a humidity of 95% ± 3% for 7 days.

#### 2.2.2. Dry–Wet Cycle Test

After the curing period, referring to the relevant requirements of unconfined compressive strength test in the Geo-technical Test Specification (SL 237-020-1999), a STWCY-1 unconfined pressure gauge (Nanjing Ningxi Soil Instrument Co., LTD, Nanjing, Jiangsu, China) is used to test the unconfined compressive strength of solidified sludge. Three parallel samples are taken from each group. During the test, the peak strength or axial compressive strain of 15% is taken as the termination condition, and the average value is taken as the unconfined compressive strength of the group of samples.

#### 2.2.3. Permeability Test

The water resistance of solidified sludge was studied by the mass loss and volume shrinkage of samples after standard curing for 7 days after drying and wetting cycles. Dry at 40 °C for 24 h and soak in distilled water for 24 h as a dry–wet cycle. After the last cycle, take the sample out of distilled water for 1 h and then weigh it. Record the mass loss of solidified sludge after different dry–wet cycles. The experimental methods were quoted by the specification for highway geotechnical Test (JTG 3430-2020).

#### 2.2.4. X-ray Diffraction Spectrum

Take 10 g from the parallel sample of unconfined compressive strength test, pass the sample through 200 mesh square sieve after low temperature blast drying at 40 °C, and use the X-ray diffraction (D8 Discover, Bruker Instruments, Ingelfingen, Germany) to analyze the phase of the screened solidified sludge.

#### 2.2.5. Pore Structure

Divide the parallel sample of unconfined compressive strength test into upper, middle and lower parts. Take 10 g blocks with a size of about 7 mm from the three parts, dry them at a low temperature blast at 40 °C for 24 h, and test the pore structure distribution of the sample with an Auto pore IV 9520(Micromeritics, Norcross, GA, UAS). The test pressure range is 25–3300 psi, and the corresponding hole diameter range is about 340 μm–6.0 nm.

#### 2.2.6. SEM

Cut a block with the size of about 10 mm from the parallel sample of unconfined compressive strength test, dry it by low temperature blast at 40 °C for 24 h, spray platinum on the surface, and observe and analyze the micro morphology of the test block by using field emission environment scanning electron microscope (FEI Quanta 3D SEM, FEI Company, Hillsboro, Oregon, American). The technical route diagram proposed for this paper is shown in Figure 1.

## 3. Results

### 3.1. Unconfined Compressive Strength

It can be seen from Table 2 and Figure 2 and Figure 3 that the strength growth rates in the table are all based on G8. The unconfined compressive strength of G8 solidified sludge without aggregate is 0.944 MPa. In general, the unconfined compressive strength of solidified sludge increases with the increase in particle size and content of building debris, and the strength growth rate after adding building debris is 8.50%~72.16%. The sample mixed with ordinary sand also has the same change trend. However, when the particle size is less than 0.25 mm, the unconfined compressive strength of solidified sludge is increased more by adding building debris. Among them, the unconfined compressive strength of 1G8S5 is 1.024 MPa, while the unconfined compressive strength of sample 1G8S’5 is 0.861. The strength ratio of both is about 1.19. When the amount of building debris is increased from 5% to 15%, the strength ratio is greater than 1 MPa. With the increase in aggregate size, the unconfined compressive strength of solidified sludge mixed with building debris is about 90% of that of ordinary sand-solidified sludge. The results show that the porous structure on the surface of small building debris adsorbs free water and reduces the thickness in the water film between sludge particles, and the role of strengthening the skeleton structure is more obvious. With the increase in particle size, the specific surface area of the skeleton material gradually decreases, the absorption of the skeleton matrix material to free water is weakened and the proportion of the advantage of the skeleton structure is increased. Compared with building debris, ordinary sand is more homogeneous and has more advantages, but the overall difference is small. It is basically feasible to use building debris as the skeleton matrix material.

Figure 2 and Figure 3 respectively, show the fitting curve of unconfined compressive strength and aggregate content of building debris (S) and ordinary sand (S’) solidified dredging sludge. It can be seen from the figure that the rule of adding building debris is basically consistent with that of ordinary sand. With the increase in aggregate size and content, its unconfined compressive strength increases linearly and rapidly, as shown in Formula (1). Except for the group of 0.075~0.25 mm, the rest R^2^ is above 0.95, which indicates that the curve fit is good.
(1)$$f_{cu}=a+bA_{w}$$
where *f_cu_* refers to unconfined compressive strength, *A_w_* refers to the amount of skeleton matrix material, and *a* and *b* are correlation coefficients.

In summary, the addition of large size building debris greatly improves the unconfined compressive strength of solidified sludge. Considering economic, applicability, green and environmental protection factors, the best research object is building debris with a particle size of 1~2 mm. When the addition amount is 10%, the unconfined compressive strength of solidified sludge for sample G8 increases by about 49%.

### 3.2. Dry Wet Cycle

It can be seen from Figure 4 that the mass loss of solidified sludge increases with the increase in the cycle period. During the test, when the dry–wet cycle lasts for one cycle, the sample shape remains basically unchanged and no obvious quality defects appear. However, after the dry–wet cycle lasts for five cycles, the test surface peels off obviously, and the mass loss rate of the solidified sludge during the dry–wet cycle that lasts for eight cycles reaches 18.43%, then the growth rate of the mass loss rate slows down with the increase in the number of cycles. The unconfined compressive strength of samples after different dry–wet cycles is shown in Figure 5. The figure shows that the unconfined compressive strength of test 4G8S10 decreases with the increase in the number of dry and wet cycles. After eight cycles of dry and wet cycles, the strength loss rate reaches 57.78%, and then the reduction in strength gradually slows down with the lengthening of the cycle. The test results show that the solidified sludge particles dehydrate during the drying and wetting cycle, resulting in the continuous increase in pores and the gradual decrease in matrix suction. With the increase in the drying and wetting cycle, when the solidified sludge shrinks to a certain extent and absorbs water again, the sludge particles are restricted by the surrounding skeleton matrix materials, and the volume is slightly expanded. At this time, the internal structure is loose and the strength is reduced.

### 3.3. Permeability

Combined with the test results of water resistance, the G8 and G8S10 groups of solidified sludge samples were selected in this experiment to study the change in permeability coefficient before and after the dry–wet cycle. The permeability coefficient of samples G8 and G8S10 is shown in Figure 6. It can be seen from the figure that the permeability coefficient of sample G8S10 is 2.00 × 10^−7^ cm/s, a permeability coefficient of 3.20 compared with sample G8 at 1 × 10^−8^ cm/s. This increase of one order of magnitude indicates that the addition of building debris increases the fine pores between solidified sludge particles, provides a seepage channel for the solidified sludge and enhances the permeability. After eight cycles of drying and wetting, the permeability coefficient of sample G8S10 has increased by one order of magnitude and the permeability coefficient of sample G8 has increased by two orders of magnitude. This increase indicates that the internal pores of sample G8 and G8S10 become larger due to the effect of drying and wetting cycles, and some of the pores between particles develop into inside particle pores, leading to an increase in the permeability coefficient. Meanwhile, the skeleton role of the skeleton material in the solidified sludge increases its resistance to drying and wetting cycles.

### 3.4. Micro-Structure

#### 3.4.1. XRD

See Figure 7 for the analysis results of X-ray diffraction patterns of samples G0, G8, G8S5, G8S10 and G8S15. The peak shape, intensity and d value of the diffraction peak in the X-ray diffraction pattern were used to identify the mineral species, and calculate the mineral content semi-quantitatively. It can be seen from Figure 7 that the main phases are illite (aluminum oxide potassium oxide silicon oxide complex), corundum (aluminum oxide), quartz (silicon dioxide), plagioclase (calcium oxide aluminum oxide silicon dioxide complex), orthoclase (calcium oxide aluminum oxide silicon dioxide complex), calcite (calcium carbonate), etc. There is also a small amount of amorphous material, but the dispersion peak is not obvious. The phase composition of sample G8, G8S5, G8S10 and G8S15 is basically similar to that of sample G0, but with slight changes. All five samples have characteristic peaks of illite, quartz (20.86°, 26.65°, 60.12°) and gismondine (8.86°, 22.05°, 51.14°). However, due to the addition of 8% curing agent to samples G8, G8S5, G8S10 and G8S15, a characteristic peak of calcium carbonate appears near 29.35°. This is because, with the increment in curing materials, the content of CaO increased, which would form calcium carbonate through carbonization reaction.

#### 3.4.2. Pore Structure

The pore size distribution of samples is shown in Figure 8. The results show that the average porosity of samples G0, G8, G8S5, G8S10 and G8S15 are 26.3%, 40.4%, 39.1%, 38.2% and 36.9%, respectively. The average pore diameters are 42.05 nm, 44.95 nm, 43.73 nm, 43.15 nm and 45.43 nm, respectively. The most probable aperture of sample G0 is 250~550 nm, and the secondary probable aperture is 30~50 nm. However, the most probable aperture of G8, G8S5, G8S10 and G8S15 is 30~50 nm, and the sub-probable aperture is 340~560 nm. It can be seen that the addition of curing agent and building debris increases the number of small holes in the sample and decreases the number of medium and large holes with a pore diameter greater than 100 nm. Combined with the analysis results of X-ray diffraction patterns, the curing agent reacts with water to generate hydrated products, forming an obvious skeleton structure with silt particles and filling harmful pores. These pores are mainly concentrated in the small holes with a pore diameter less than 100 nm. The most probable pore size of G8S10 and G8S15 solidified dredged sludge is lower than that of sample G8, indicating that slightly large pores between skeletons are filled with building debris.

#### 3.4.3. SEM

The SEM images of samples G0 and G8S10 at 3000 times magnification show that a large amount of Ca^2+^ has been introduced into the curing agent, as shown in Figure 9. Since the curing method involves wrapping with plastic wrap, calcium hydroxide (C-H) and a small amount of other hydration products (such as C-S-H, AFt, etc.) are formed in the solidified soil. These products fill the gaps between loose clay particles and cement them together to form a “hydration product clay” agglomerated cementation structure. However, this structure contributes little to the filling of silt soil pores, and only slightly increases the strength. However, skeleton matrix materials such as building debris have greatly contributed to the filling of larger pore sizes, making the solidified soil more dense and providing greater rigidity for the solidified aggregate structure. This improves the overall particle size distribution and structure, and the introduction of skeleton materials has increased the strength of the solidified soil by 49%.

## 4. Discussion

The addition of construction waste debris greatly improves the strength of solidified dredged sludge, mainly due to two reasons. Firstly, after the addition of cement, fly ash, sodium silicate and other materials, a hydration reaction occurs, generating more C-H, C-S-H gel and a small amount of hydrated calcium silicate and other hydration products [12]. This reaction plays a dual role of filling and cementation, improving the strength of solidified soil to a certain extent, but also adds more micro-pores, so the strength contribution rate is slightly lower. Secondly, the introduction of skeleton matrix materials, such as building debris, further fills pores with slightly larger diameters, reducing the porosity of the solidified soil. Building debris is a rigid material with a large elastic and a high hardness and it is not easy to deform. Therefore, adding rigid materials to the solidified sludge is helpful. Figure 10 shows that *σ*_1_ is the principal stress of silt, *σ*_2_ is the principal stress of agglomerate cement and *σ*_3_ is the principal stress of building debris, with *σ*_1_ < *σ*_2_ < *σ*_3_. The average principal stress of a unit section is determined by σ_1_. The reinforcement is $\frac{\sum \sigma_{1i}+\sum \sigma_{2j}+\sum \sigma_{3k}}{i+j+k}$. This shows that the curing agent and building debris have a synergistic effect on the overall structural strength. Building debris played an important role in the improving the unconfined compressive strength. It mainly consists of silicon raw materials such as quartz, basalt, etc. The powders in building debris filled the pores of solidified silt, improving its compaction.

## 5. Conclusions

In this paper, a skeleton matrix, such as construction waste debris, is introduced to modify solidified dredged sludge. The optimal scheme of construction waste debris used for modified solidified sludge is discussed through unconfined compressive strength, X-ray diffraction pattern, mercury intrusion analysis and scanning electron microscope image analysis. The relevant conclusions are as follows:(1)When mixing construction waste debris, ordinary sand and other skeleton materials into the solidified, dredged sludge material, the unconfined compressive strength increases with the increase in particle size and content of the skeleton material. The unconfined compressive strength of solidified sludge with building waste debris ranges from 89% to 119% of that of ordinary sand solidified sludge, while the water stability of solidified sludge is improved. It indicates that construction waste debris has a significant reinforcement effect on the modification of solidified silt.(2)Due to the addition of a curing agent, calcium hydroxide (C-H) and a small amount of other hydration products (such as C-S-H), loose clay particles are cemented together, most of the macro pores are filled, and a particle skeleton structure is formed in the soil mass. The addition of building debris further improves the pore structure of solidified sludge, provides a rigid aggregate support system and greatly improves the principal stress of the unit section of solidified sludge, thus greatly improving the strength of solidified sludge.(3)Considering economic applicability, environmental protection and other factors, it is recommended that building debris with a particle size of 1~2 mm and a content of about 10% be used as the modified proportion of solidified dredged sludge materials in future engineering applications to improve the water stability of solidified sludge. Large-scale use of construction waste can also reduce the disposal cost of solidified soil and a new method for the disposal of river sludge can be provided in future.

## Figures and Tables

**Figure 1 materials-16-02780-f001:**
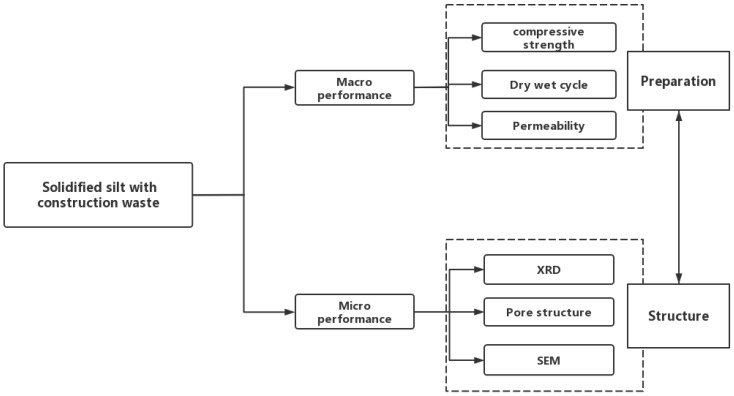
Technical route diagram.

**Figure 2 materials-16-02780-f002:**
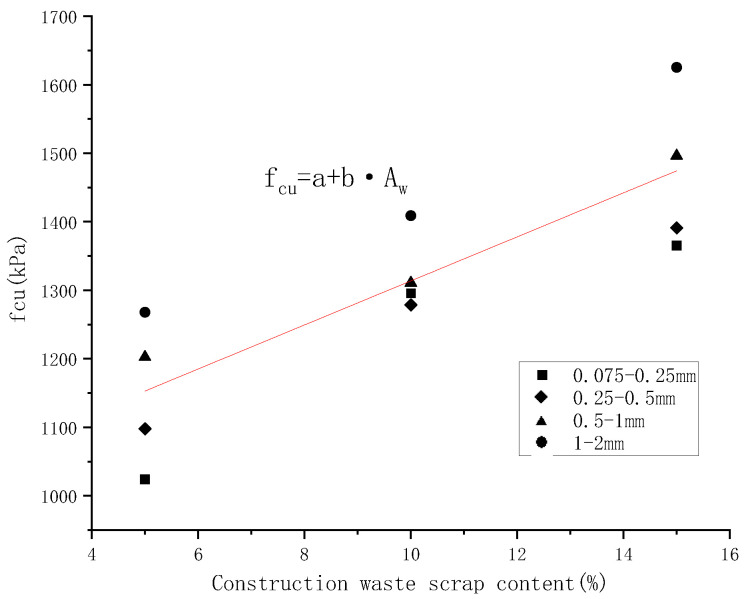
Relationship between solidified building debris silt’s unconfined compressive strength with content.

**Figure 3 materials-16-02780-f003:**
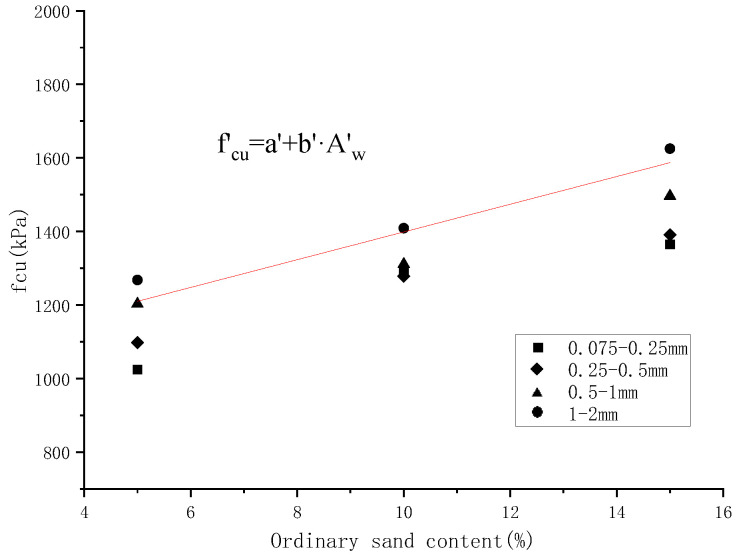
Relationship between solidified ordinary sand silt’s unconfined compressive strength with content.

**Figure 4 materials-16-02780-f004:**
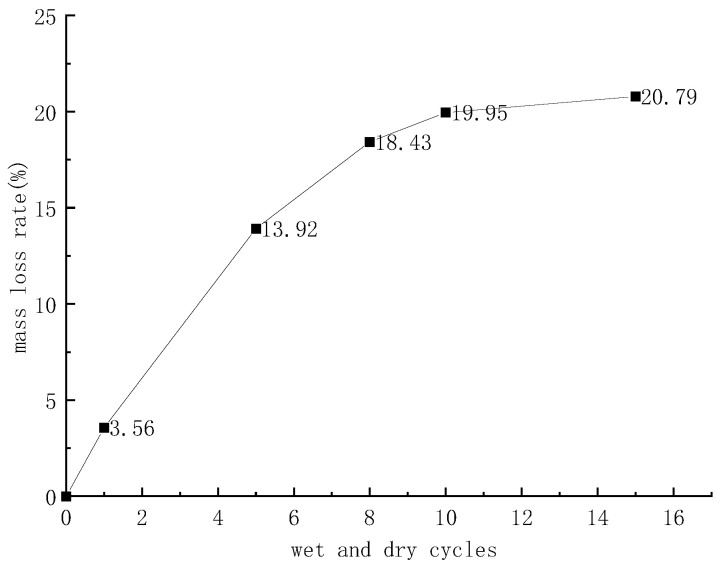
Mass loss rate of 4G8S10 under different wet and dry cycles.

**Figure 5 materials-16-02780-f005:**
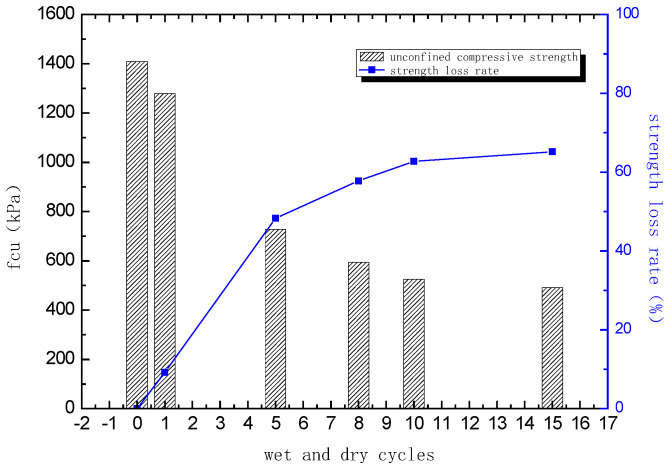
Intensity variation in 4G8S10 under different wet and dry cycles.

**Figure 6 materials-16-02780-f006:**
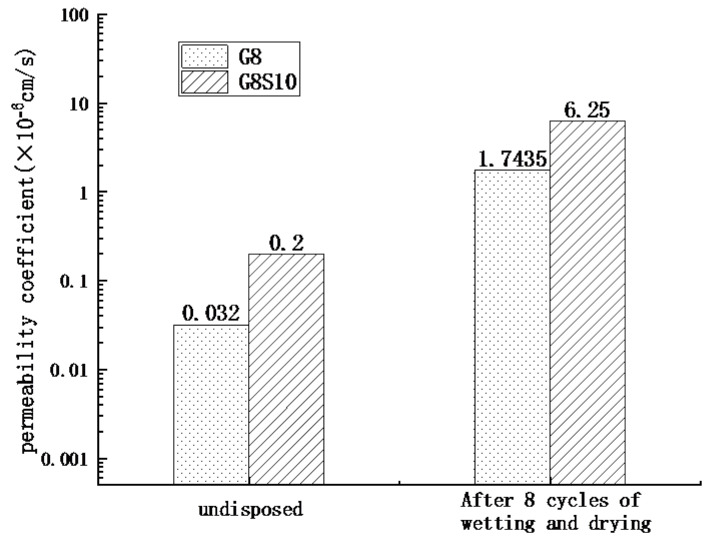
Permeability coefficient of the sample before and after the wetting and drying cycle.

**Figure 7 materials-16-02780-f007:**
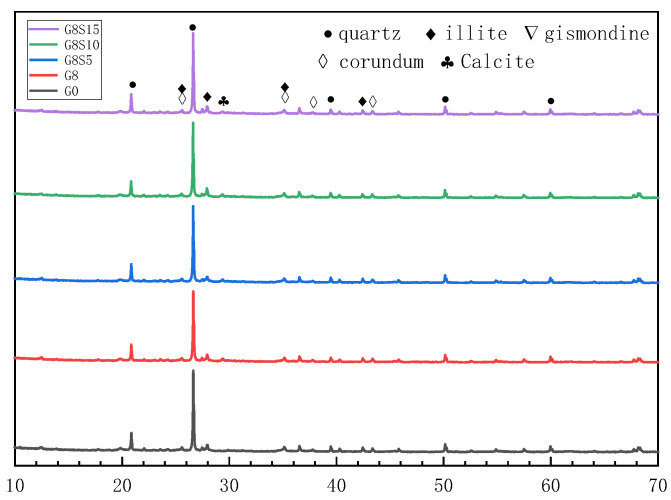
X-ray diffraction patterns of G0, G8, G8S5, G8S10 and G8S15.

**Figure 8 materials-16-02780-f008:**
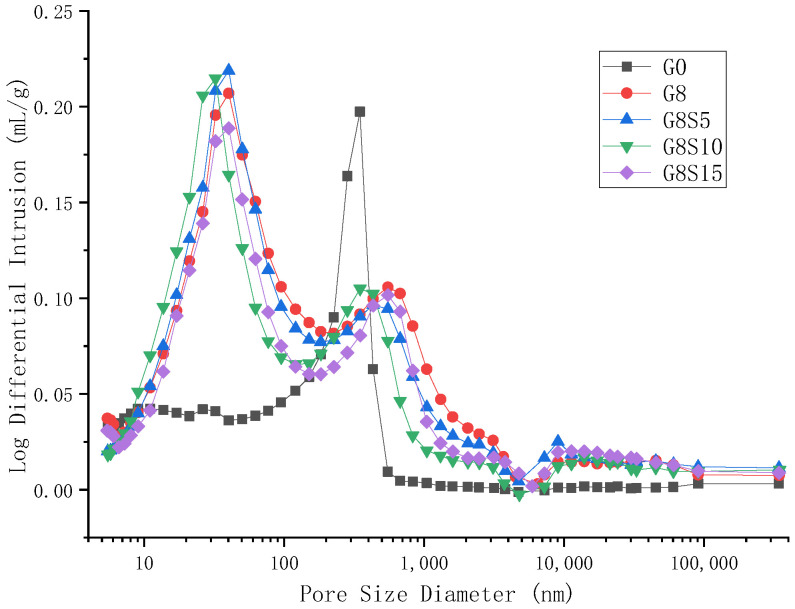
Pore size distribution of sample G0, G8, G8S5, G8S10 and G8S15.

**Figure 9 materials-16-02780-f009:**
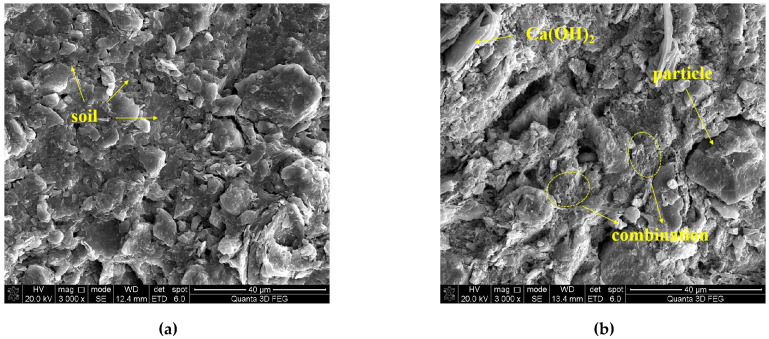
SEM images of G0 and G8S10. (**a**) G0; (**b**) G8S10.

**Figure 10 materials-16-02780-f010:**
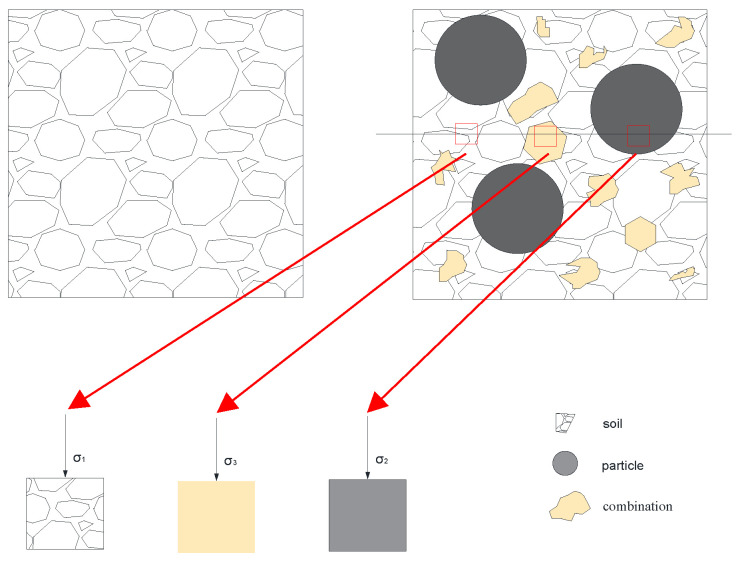
Schematic diagram of unit section’s principal stress after silt solidification.

**Table 1 materials-16-02780-t001:** Ratio of solidified silt.

Building Debris (S)			Ordinary Sand (S’)		
Serial Number	Particle Size	Mixing Amount (%)	Serial Number	Particle Size	Mixing Amount (%)
1G8S5	0.075~0.25 mm	5	1G8S’5	0.075~0.25 mm	5
1G8S10		10	1G8S’10		10
1G8S15		15	1G8S’15		15
2G8S5	0.25~0.5 mm	5	2G8S’5	0.25~0.5 mm	5
2G8S10		10	2G8S’10		10
2G8S15		15	2G8S’15		15
3G8S5	0.5~1 mm	5	3G8S’5	0.5~1 mm	5
3G8S10		10	3G8S’10		10
3G8S15		15	3G8S’15		15
4G8S5	1~2 mm	5	4G8S’5	1~2 mm	5
4G8S10		10	4G8S’10		10
4G8S15		15	4G8S’15		15

**Table 2 materials-16-02780-t002:** Compressive strength between building debris (S) and ordinary sand (S’) solidified soil.

Building Debris (S)			Ordinary Sand (S’)			f_cu-S_/f_cu-S’_
Serial Number	f_cu-S_ (MPa)	Growth of f_cu_ (%)	Serial Number	f_cu-S’_ (MPa)	Growth of f_cu_ (%)	
G8	0.944	0.00%	G8	0.944	0.00%	/
1G8S5	1.024	8.50%	1G8S’5	0.861	−8.80%	1.19
1G8S10	1.296	37.25%	1G8S’10	1.262	33.72%	1.03
1G8S15	1.365	44.62%	1G8S’15	1.352	43.17%	1.01
2G8S5	1.098	16.31%	2G8S’5	1.189	25.97%	0.92
2G8S10	1.279	35.43%	2G8S’10	1.359	43.97%	0.94
2G8S15	1.391	47.37%	2G8S’15	1.509	59.81%	0.92
3G8S5	1.203	27.38%	3G8S’5	1.337	41.58%	0.90
3G8S10	1.311	38.86%	3G8S’10	1.456	54.24%	0.90
3G8S15	1.496	58.50%	3G8S’15	1.640	73.76%	0.91
4G8S5	1.268	34.32%	4G8S’5	1.430	51.52%	0.89
4G8S10	1.409	49.25%	4G8S’10	1.563	65.61%	0.90
4G8S15	1.625	72.16%	4G8S’15	1.825	93.29%	0.89

## Data Availability

Not applicable.

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
