# Peer review of "Study on Modification and Mechanism of Construction Waste to Solidified Silt"

_materials, 2023, doi:10.3390/ma16072780_

Round 1

Reviewer 1 Report

The manuscript entitled “Study on the mechanism of modification of construction waste to solidified silt” presented experimental study. The influence of different parameters was studied and analyzed. However, there seems to be little to no novelty in this study.

This reviewer recommends major editing and resubmits for re-review.

Comments:

  • The English writing of the manuscript needs improvement. Therefore, it could benefit greatly from professional editing to improve technical writing and English.
  • Please mention your study limits and suggest some future research topics
  • In References, the sources are written in different styles. Please update the reference list.  It is necessary to bring in accordance with the requirements of the magazine for the design of References. If possible, indicate DOI.
  • Please use some innovative keywords.
  • Please mention your study limits in the abstract.
  • The literature can be expanded by studying some of these papers.
    • Predictive modeling of swell-strength of expansive soils using artificial intelligence approaches: ANN, ANFIS and GEP
    • Predictive modeling for sustainable high-performance concrete from industrial wastes: A comparison and optimization of models using ensemble learners
    • Promoting the green Construction: Scientometric review on the mechanical and structural performance of geopolymer concrete
  • The Conclusions should reflect what the practical application of the results obtained in this study is. In what climatic conditions should the recommendations of the authors be taken into account?
  • The authors should increase their discussion on previous related research and highlight how their study is providing a different approach or adding significantly to what has been done. The authors have to explain what is the new here in comparison with the previous studies. The novelty of the current work should be highlighted in the introduction. Please try to mention a problem that needs solving - in other words, the research question underlying your study clearer.
  • The title of the manuscript should be revised.
  • Some types of standards should be used to perform different experimental studies. Please provide details for the standards used in each study.
  • Section 4 should be discussed in detail.
  • The authors must redo the Abstract and bring it in compliance with the requirements of the journal. The scientific problem is poorly described (Background). The scientific novelty is not indicated. I recommend shortening the Abstract to 200 words. Editors strongly encourage authors to use the following style of structured abstracts, but without headings: (1) Background: Place the question addressed in a broad context and highlight the purpose of the study; (2) Methods: Briefly describe the main methods or treatments applied; (3) Results: Summarize the article's main findings; and (4) Conclusions: Indicate the main conclusions or interpretations. The abstract should be an objective representation of the article
  • It is advisable to add a flowchart at the beginning of the paper. Then the article would become more visual and structured
  • The economic aspects are also required for sustainability in social aspect. It is suggested to authors to evaluate the cost-benefit study of this as a further investigation
  • The conclusion should be an objective summary of the most important findings in response to the specific research question or hypothesis. A good conclusion states the principal topic, key arguments and counterpoint, and might suggest future research. It is important to understand the methodological robustness of your study design and report your findings accordingly. Please improve your conclusion section.

Author Response

Dear Reviewer,

I would like to thank you for taking the time and effort to go through my paper and providing constructive criticisms which are extremely valuable for me. I appreciate your thoughtful review and am grateful for your valuable insights.

I believe those changes have significantly improved the paper and I thank you again for your careful review. I have made all the suggested changes, and I hope my paper meets your approval.

Point 1: The English writing of the manuscript needs improvement. Therefore, it could benefit greatly from professional editing to improve technical writing and English.

Response 1: Thank you for the suggestion, the manuscript have been checked by native English researcher. The authors can provide proof of polishing.

Point 2: Please mention your study limits and suggest some future research topics

Response 2: Thank you for the suggestion, we have added these in introduction.

Point 3: In References, the sources are written in different styles. Please update the reference list.  It is necessary to bring in accordance with the requirements of the magazine for the design of References. If possible, indicate DOI.

Response 3:  Thanks for the suggestion, the reviewer is correct,  we have added all references DOI, and cited more references[20-22] according to reviewer’s suggestions.

Point 4: Please use some innovative keywords.

Response 4: Thanks for the suggestion, We have revised keywords.

Point 5: Please mention your study limits in the abstract.

Response 5: Thanks for the suggestion, We have revised abstract.

Point 6:The literature can be expanded by studying some of these papers.

Predictive modeling of swell-strength of expansive soils using artificial intelligence approaches: ANN, ANFIS and GEP

Predictive modeling for sustainable high-performance concrete from industrial wastes: A comparison and optimization of models using ensemble learners

Promoting the green Construction: Scientometric review on the mechanical and structural performance of geopolymer concrete

Response 6:  Thanks for the suggestion, the reviewer is correct,  we have cited more references[20-22] according to reviewer’s suggestions.

Point 7:The Conclusions should reflect what the practical application of the results obtained in this study is. In what climatic conditions should the recommendations of the authors be taken into account?

Response 7: Thanks for the suggestion, We have improve conclusion as shown in line 285-309.

Point 8: The authors should increase their discussion on previous related research and highlight how their study is providing a different approach or adding significantly to what has been done. The authors have to explain what is the new here in comparison with the previous studies. The novelty of the current work should be highlighted in the introduction. Please try to mention a problem that needs solving - in other words, the research question underlying your study clearer.

Response 8:  Thanks for the suggestion, the authors have  revised introduction, we found that large amount of silt may produce in river and lake regulation, it not only occupies land, but also pollutes the environment, so it is urgent to seek effective disposal and utilization ways. Present studies all aimed at how to modified the solidified clay, some researchers chose steel waste slag, slag and other wastes to modified solidified clay and the solidified silt has high strength and water resistance. However, the properties of solid waste materials are uneven, the performance is unstable, and the chemical and physical properties of river sludge vary greatly. How to use them more efficiently is still a current problem. This paper proposed a conclusion that In consideration of economic applicability, environmental protection and other factors, it is recommended that building debris with a particle size of 1~2 mm and a content of about 10% be used as the modified proportion of solidified dredged sludge materials in future engineering applications to improve the water stability of solidified sludge.

Point 9:The title of the manuscript should be revised.

Response 9: Thanks for the suggestion, after careful consideration of all authors, the title of the manuscript has  well summarized all research content, so hold the title temporarily.

Point 10: Some types of standards should be used to perform different experimental studies. Please provide details for the standards used in each study.

Response 10: Thanks for the suggestion,  we  quoted the  experimental standards such as the heavy compaction test in Soil Test Specification (SL 237-1999), Geo-technical Test Specification (SL 237-020-1999), the specification for highway geotechnical Test (JTG 3430-2020) and so on.

Point 11: Section 4 should be discussed in detail.

Response 11: Thanks for the suggestion, We have revised discussions.

Point 12: The authors must redo the Abstract and bring it in compliance with the requirements of the journal. The scientific problem is poorly described (Background). The scientific novelty is not indicated. I recommend shortening the Abstract to 200 words. Editors strongly encourage authors to use the following style of structured abstracts, but without headings: (1) Background: Place the question addressed in a broad context and highlight the purpose of the study; (2) Methods: Briefly describe the main methods or treatments applied; (3) Results: Summarize the article's main findings; and (4) Conclusions: Indicate the main conclusions or interpretations. The abstract should be an objective representation of the article

Response 12: Thanks for the suggestion, We have revised abstract.

Point 13:   It is advisable to add a flowchart at the beginning of the paper. Then the article would become more visual and structured

Response 13: Thanks for the suggestion, we have added Figure 1.

Point 14:   The economic aspects are also required for sustainability in social aspect. It is suggested to authors to evaluate the cost-benefit study of this as a further investigation

Response 14: Thanks for the suggestion, We have revised conclusion and provide a  a vision of economic aspects for future.

Point 15: The conclusion should be an objective summary of the most important findings in response to the specific research question or hypothesis. A good conclusion states the principal topic, key arguments and counterpoint, and might suggest future research. It is important to understand the methodological robustness of your study design and report your findings accordingly. Please improve your conclusion section.

Response 15: Thanks for the suggestion, We have improve conclusion as shown in line 285-309.

Reviewer 2 Report

The manuscript has been drafted quite well. Each section has been prepared, explained and discussed with great care by the researchers. I have few minor suggestions that should be addressed before the final draft for better understanding of the readers:

·         The length of the sentences at various places is too long making it difficult to understand (example: Page 2, Line 53-63 is a single huge sentence). It is advised to go through the manuscript and reduces the sentence length wherever possible.

·         The researchers are required to explain the reason for the generation of characteristic peak of calcium carbonate at 29.35 ° with the addition of 8% curing agent.

Author Response

Dear Reviewer,

I would like to thank you for taking the time and effort to go through my paper and providing constructive criticisms which are extremely valuable for me. I appreciate your thoughtful review and am grateful for your valuable insights.

I believe those changes have significantly improved the paper and I thank you again for your careful review. I have made all the suggested changes, and I hope my paper meets your approval.

Point 1:  the length of the sentences at various places is too long making it difficult to understand (example: Page 2, Line 53-63 is a single huge sentence). It is advised to go through the manuscript and reduces the sentence length wherever possible.

Response 1: Thanks for your attention, we have revised it as “ Through controlling the amount of admixture and particle size, the strength of the solidified sludge and  solidification effect will be adjusted. X-ray diffraction, and scanning electron microscopy were used to analyze the physical phase, pore structure and micro morphology of the solidified sludge. Then, it will reveal the mechanism of action and solidification of construction waste debris in the process of solidification of dredged sludge, and screen out the better scheme. Finally, a theoretical basis for the application of solidified soil in water conservancy, municipal, construction and other industrial projects will be provided. It is therefore recommended that the construction waste debris can reliably be used for modifying solidified silt to improve stability and the compaction characteristics.” in line 66-76.

Point 2: The researchers are required to explain the reason for the generation of characteristic peak of calcium carbonate at 29.35 ° with the addition of 8% curing agent.

Response 2: The reviewer is correct, we have revised it as “It was because with the increment of curing materials,  the content of CaO increased, which would form calcium carbonate for carbonization reaction.” in line 235-236.

Round 2

Reviewer 1 Report

Accept